# The Entero-Mammary Pathway and Perinatal Transmission of Gut Microbiota and SARS-CoV-2

**DOI:** 10.3390/ijms231810306

**Published:** 2022-09-07

**Authors:** Carmen Josefina Juárez-Castelán, Juan Manuel Vélez-Ixta, Karina Corona-Cervantes, Alberto Piña-Escobedo, Yair Cruz-Narváez, Alejandro Hinojosa-Velasco, María Esther Landero-Montes-de-Oca, Eduardo Davila-Gonzalez, Eduardo González-del-Olmo, Fernando Bastida-Gonzalez, Paola Berenice Zárate-Segura, Jaime García-Mena

**Affiliations:** 1Departamento de Genética y Biología Molecular, Cinvestav, Av. Instituto Politécnico Nacional 2508, Ciudad de México 07360, Mexico; 2Laboratorio de Posgrado de Operaciones Unitarias, Escuela Superior de Ingeniería Química e Industrias Extractivas, Instituto Politécnico Nacional, Ciudad de México 07738, Mexico; 3División de Neonatología, Hospital de Ginecología y Obstetricia, Instituto Materno Infantil del Estado de México, Toluca de Lerdo 50170, Mexico; 4Departamento de Ginecología y Obstetricia, Hospital Gustavo Baz Prada, ISEM, Nezahualcóyotl 57300, Mexico; 5Laboratorio Estatal de Salud Pública del Estado de México, ISEM, Toluca de Lerdo 50180, Mexico; 6Laboratorio de Medicina Traslacional, Escuela Superior de Medicina, Instituto Politécnico Nacional, Ciudad de México 11340, Mexico

**Keywords:** SARS-CoV-2, mother rectal swab, neonate rectal swab, human colostrum, high-throughput DNA sequencing, fecal microbiota, RT-ddPCR, RT-qPCR, gut microbiota

## Abstract

COVID-19 is a severe respiratory disease threatening pregnant women, which increases the possibility of adverse pregnancy outcomes. Several recent studies have demonstrated the ability of SARS-CoV-2 to infect the mother enterocytes, disturbing the gut microbiota diversity. The aim of this study was to characterize the entero-mammary microbiota of women in the presence of the virus during delivery. Fifty mother–neonate pairs were included in a transversal descriptive work. The presence of SARS-CoV-2 RNA was detected in nasopharyngeal, mother rectal swabs (MRS) and neonate rectal swabs (NRS) collected from the pairs, and human colostrum (HC) samples collected from mothers. The microbiota diversity was characterized by high-throughput DNA sequencing of V3-16S rRNA gene libraries prepared from HC, MRS, and NRS. Data were analyzed with QIIME2 and R. Our results indicate that several bacterial taxa are highly abundant in MRS positive for SARS-CoV-2 RNA. These bacteria mostly belong to the Firmicutes phylum; for instance, the families Bifidobacteriaceae, Oscillospiraceae, and Microbacteriaceae have been previously associated with anti-inflammatory effects, which could explain the capability of women to overcome the infection. All samples, both positive and negative for SARS-CoV-2, featured a high abundance of the Firmicutes phylum. Further data analysis showed that nearly 20% of the bacterial diversity found in HC was also identified in MRS. Spearman correlation analysis highlighted that some genera of the Proteobacteria and Actinobacteria phyla were negatively correlated with MRS and NRS (*p* < 0.005). This study provides new insights into the gut microbiota of pregnant women and their potential association with a better outcome during SARS-CoV-2 infection.

## 1. Introduction

The actual well-known Coronavirus disease 2019 (COVID-19) is produced by the severe acute respiratory syndrome coronavirus 2 (SARS-CoV-2), causing a worldwide pandemic that began in 2020. At the population level, pregnant women are a vulnerable group, with a greater risk of COVID-19 morbidity and mortality, increasing the risk of adverse pregnancy outcomes [1]. In this context, statistics from the UK reported a total of 9% of pregnant women and 6-week postpartum COVID-19 admissions to intensive care units (ICU) [2], while a review highlighted that most infections occurred during the third trimester, 11% of pregnant women with COVID-19 required admission to ICU, and 8% required mechanical ventilation [3]. It has also been reported that prevalence among women is variable, rising from 0.5% to 5% in the span of 2 weeks [4]. Regarding the Mexican population, it was reported that during the second peak of the pandemic, approximately 12% of asymptomatic pregnant women were positive for the virus [1].

Recently, several studies have demonstrated the ability of SARS-CoV-2 to infect and replicate in enterocytes of the human small intestine, in addition to RNA virus detection in fecal samples and the altered gut microbiota structure in SARS-CoV-2 positive patients [5,6]. It is known that SARS-CoV-2 infection is related to the angiotensin-converting enzyme receptor 2 (ACE2), whose activity is influenced and, in turn, affects the functionality of the gut microbiota [7]. For this reason, the association between gut microbiota and SARS-CoV-2 infection has been studied, given that the gut microbiota not only supports mucosal immunity but also modulates the systemic immune response in the host [8]. Reduction in the abundance of key immunomodulatory species of the gut microbiota, such as Bifidobacteria, *Faecalibacterium prausnitzii*, or *Eubacterium rectale*, has been reported in COVID-19 patients [5,9].

The alteration in the gut microbiota of pregnant women is an important aspect of the study since the maternal gut microbiota plays a crucial role in the first microbiota and immunity of the newborn. It has been reported that the maternal microbiota induces the activation of B cells [10,11] and other immunological components that migrate to the mammary gland to be transferred to the neonate through human milk [12]. Moreover, there is an interesting hypothesis for the translocation of internal bacteria from the mother’s gastrointestinal (GI) tract into the mammary gland, driven by immune cells and occurring during the late stages of pregnancy. In this manner, the maternal GI tract is a source of bacteria for the milk microbiota, and the maternal gut microbiota is vertically transferred to the infant via human milk [13,14]. Human milk contains specific microbiota comprising mostly bacteria such as *Staphylococcus*, *Pseudomonas*, *Streptococcus*, *Propionibacterium*, and *Sphingomonas*, important anaerobic bacteria such as *Bifidobacterium*, *Faecalibacterium*, and *Akkermansia*, and remarkable lactic acid bacteria such as *Lactobacillus* [15,16]. This microbiota is the source of the first colonizing bacteria of the neonate’s GI tract, playing important roles in the metabolism of milk sugar, influencing the development of the immune system, producing secondary metabolites such as short-chain fatty acids (SCFA), vitamins, contributing to the reduction of infections by potential pathogens, producing antimicrobial components, and improving the function of the neonate’s intestinal barrier [17].

Despite the important role of maternal gut microbiota in the human milk microbiota and neonatal health, limited information has been published with respect to the COVID-19 influence on the maternal microbiota or the effect of SARS-CoV-2 maternal infection on the human milk bacterial diversity and microbial populations. The aim of this work was to analyze the microbiota from rectal swab samples, taken during delivery, of women with detectable presence of SARS-CoV-2 genome RNA. The diversity of the bacterial microbiota found in their human colostrum and in rectal swab samples from their newborns was also characterized to compare with the bacterial community found in samples from healthy women. The results obtained in this study provide interesting insight into changes occurring in bacterial community diversity in the infection state and its influence on newborn health.

We hypothesize that an alteration of the gut microbiota profile during SARS-CoV-2 infection in the mother is associated with changes in the bacterial community in the human colostrum, which potentially will influence bacterial gut colonization in the newborn.

## 2. Results

### 2.1. Presence of SARS-CoV-2 Genomic RNA in the Mother–Neonate Pairs

The sampled women were on average 25 years old, with an age range from 16 to 38 years, while the gestational age revealed an average of approximately 39 weeks. Their heights and weights were within common ranges for women in the Mexican population, as well as the BMI data, which indicated a high abundance of overweight and obesity conditions in the sample (Table 1). The 50 neonates included in the study were less than 6 days old, 32% of them were female, and 68% were male, with an average weight of 2.92 (±0.42) kg and size of 49.51 (±1.47) cm. The mother rectal swab (MRS), human colostrum (HC), and neonate rectal swab (NRS) samples of the 50 mother–neonate pairs were initially characterized for the presence of SARS-CoV-2 genome RNA using RT-ddPCR to detect the N gene. The RT-ddPCR test results were positive for two-thirds of mothers at the time of delivery, whereas 44 of them (88%) had no COVID-19 symptoms (Table 1). The RT-qPCR results for detection of SARS-CoV-2 in nasopharyngeal swabs of the same mothers were positive for nearly one-third of the RT-ddPCR positive women, while 100% of the RT-ddPCR negative women were also negative based on RT-qPCR. No other data collected from the same women, such as blood tests, risk factors, parity, or socioeconomic data, showed any interesting association or tendency with the presence of SARS-CoV-2 RNA in MRS (Table 1).

For the HC samples, less than half of the women expressed positive results for the virus using RT-ddPCR (Table 1). On the other hand, SARS-CoV-2 genome RNA detection by digital PCR in NRS showed that half of neonates carried the viral genome. Interestingly among the positive cases, only one-fifth were positive for the presence of the virus by RT-qPCR detection of SARS-CoV-2 in nasopharyngeal swabs (Table 2). No other collected variable in the neonates, such as the qualification status at birth and the somatometric data, showed any interesting association or tendency with the presence of SARS-CoV-2 RNA in MRS (Table 2).

### 2.2. Diversity of Microbiota Taxa and SARS-CoV-2 Positivity

The microbiota diversity in MRS, HC, and NRS samples was characterized by semiconductor sequencing of V3-16S rRNA gene libraries. The total reads had an average of 2 million for each type of sample, with a median quality score of 30 (Table 3). The rarefaction plots also indicated satisfactory sequencing (Appendix A). At the phylum level, the microbiota diversity showed a predominance of Bacteroidetes and Firmicutes phyla in the MRS samples; a large relative abundance of Firmicutes phylum in HC, and a dominance of Firmicutes and Proteobacteria phyla in the NRS samples (Figure 1A). There was no statistically significant difference in the phyla’s relative abundances associated with the detection of SARS-CoV-2 by RT-ddPCR (Appendix A). At the family level, for the MRS samples, the bacterial diversity was characterized by a predominance of the Prevotellaceae (Bacteroidetes), Ruminococcaceae (Firmicutes), and Enterobacteriaceae (Proteobacteria) families (Figure 1B). However, only Bifidobacteriaceae (Actinobacteria) and Oscillospiraceae (Firmicutes) were more abundant in the SARS-CoV-2 positive cases by digital PCR, while Microbacteriaceae (Actinobacteria) was more abundant in the negative cases with statistical significance (Appendix A). The diversity at the family level in the HC samples showed a clear predominance of Staphylococcaceae (Firmicutes) in both positive and negative samples for SARS-CoV-2 by digital PCR (Figure 1B), with no statistical significance (Appendix A). On the other hand, the family abundance in the NRS samples showed a predominance of Streptococcaceae (Firmicutes) and Enterobacteriaceae in the negative samples for SARS-CoV-2 by RT-ddPCR, and a predominance of Enterobacteriaceae in the positive samples (Figure 1B). In this case, the comparatively higher abundance of Streptococcaceae (Firmicutes) in the negative, and higher abundance of Ruminococcaceae in the positive, samples for SARS-CoV-2 by RT-ddPCR were statistically significant (Appendix A).

### 2.3. Alfa and Beta Diversity of the Bacterial Microbiota and SARS-CoV-2 Positivity

The analyses characterizing the alfa diversity of MRS, HC, and NRS samples did not show statistically significant difference for observed number of species, Chao 1, Shannon, and Simpson indexes (Appendix A); there was only a slight tendency for higher observed number of species in the MRS, HC, and NRS positive samples for SARS-CoV-2 by RT-ddPCR (Figure 2A). The beta diversity, on the other hand, clustered apart the MRS from the HC samples with statistical significance, regardless of the SARS-CoV-2 detection by RT-ddPCR (Figure 2B). For the MRS samples, there was statistically significant difference between the beta diversity associated with the detection of SARS-CoV-2 by RT-ddPCR (Appendix A).

### 2.4. SARS-CoV-2 Positivity Is Associated with an Increased Abundance of Several Bacterial Taxa

The LEfSe analysis of the bacterial community present in the MRS, HC, and NRS samples revealed bacterial taxa with large relative abundances associated with the presence of the SARS-CoV-2 genome as detected by RT-ddPCR. The negative MRS samples had an increased abundance of bacterial members of the phyla Actinobacteria, Bacteroidetes, Firmicutes, and Proteobacteria, while the positive samples exhibited an increased abundance of the same phyla plus members of the phylum Verrucomicrobiota (Figure 3A). Remarkably, there was an increase in the abundance of bacteria of genera such as *Akkermansia*, *Butyricimonas*, and families Lachnospiraceae and Ruminococcaceae associated with the detection of SARS-CoV-2. The negative HC samples showed an increase in the abundance of bacterial members of the Actinobacteria and Firmicutes phyla, while the positive HC samples had an increase in bacteria from the same phyla plus Cyanobacteria and Proteobacteria (Figure 3B). In this case there was an increase in the abundance of bacteria of genera such as *Lawsonella* and families such as *Ruminoccocaceae* associated with SARS-CoV-2. Finally, the NRS negative samples showed an increase in the abundance of bacterial taxa of the Actinobacteria, Bacteroidetes, Firmicutes, and Proteobacteria phyla, whereas the NRS positive samples included only Actinobacteria, Firmicutes, and Proteobacteria (Figure 3C). There was an increase in the genera *Bifidobacterium*, *Coprococcus*, and *Phascolarctobacterium* associated with SARS-CoV-2 detection. The change in the abundance of all mentioned taxa was statistically significant (Appendix A).

### 2.5. Shared Taxa and Differential Frequency of Taxa among Samples

An analytical inspection of the feature table obtained for each type of sample showed 102 bacterial taxa shared between MRS and HC; 162 taxa shared between HC and NRS, and 168 bacterial taxa shared between MRS and NRS samples (Figure 4A). The analysis of high-throughput sequencing data for the microbiota based on the negative binomial distribution showed that in the MRS SARS-CoV-2 positive samples by RT-ddPCR, there was an increase in the abundance of members of Firmicutes such as *Clostridia_UCG-014*, *Megasphaera*, *Ruminococcaceae_NK4A214_group*, *Ruminococcus*, *Ruminococcaceae_UCG-005*, *Ruminococcaceae_Uncultured*, *Christensenellaceae*_R−7_group, *Erysipelotrichaceae_UCG-003*, *Roseburia*, *Ruminococcaceae*_*UCG-002*; phylum Actinobacteriota such as *Bifidobacterium*; phylum Verrucomicrobiota such as *Akkermansia*, and phylum Bacteroidetes such as *Bacteroides*, and a decrease in the abundance of Actinobacteriota, genus *Pseudonocardia*. For the MRS SARS-CoV-2 negative samples by RT-ddPCR, the increase was for Firmicutes such as *Ezakiella* and Bacteroidetes such as *Porphyromona*s (Figure 4B). The NRS SARS-CoV-2 positive samples had an increase in the abundance of the genus *Blautia* in Firmicutes, while the genera *Streptococcus* (Firmicutes) and *Mesorhizobium* (Proteobacteria) were increased in the MRS SARS-CoV-2 negative samples (Figure 4B). On the other hand, only the HC samples showed a decrease in the Firmicutes *Blautia* by this analysis (Figure 4B).

### 2.6. Origin of Bacterial Taxa Present in the Human Colostrum Samples

The potential origin of bacteria present in the HC was explored. SourceTracker analysis estimated that approximately 21% of the identified bacteria in the HC had a common source with the bacteria characterized in MRS, while the remaining 79% came from unknown sources (*p* < 0.006, Wilcoxon signed-rank test) (Figure 5A). The relative abundance of bacteria found in HC classified by the analysis of MRS origin, showed that members of phyla Firmicutes *Staphylococcus* (46%), and *Streptococcus* (6%), Proteobacteria *Escherichia* (11%), Actinobacteria *Microbacterium* (7%), and Bacteroidetes *Prevotella* (5%) were more abundant, while the relative abundance of taxa in the Unknown source was dominated by the Firmicutes *Staphylococcus* (47%), *Lactobacillus* (4%) and Actinobacteria *Cutibacterium* (5%) (Figure 5B and Appendix A).

### 2.7. Bacterial Taxa Correlating with the Presence of SARS-CoV-2 and Metadata Collected in Mothers

The correlation of the bacterial taxa abundance with numerical clinical metadata was explored. The Spearman analyses showed that in the MRS samples with a significance of *p* < 0.001, members of bacterial taxa such as Firmicutes (*Colidextribacter*, Lachnospiraceae, and *Flavonifractor*) were positively correlated with the presence of SARS-CoV-2 as detected by RT-ddPCR, while *Lactococcus* (Firmicutes) was negatively correlated with the presence of SARS-CoV-2. On the other hand, the SARS-CoV-2 virus genomic RNA detection by RT-qPCR in nasopharyngeal swabs collected from the mothers was positively correlated with members of the phyla Proteobacteria (*Undibacterium*) and Firmicutes (Oscillospiraceae family and *Faecalibacterium UBA1819*), and negatively correlated with Proteobacteria (Beijerrinckiaceae and *Bradyrhizobium*), Actinobacteria (*Slackia* and *Microbacterium*), Firmicutes (*Fenollaria*), and Proteobacteria (*Sphingomonas* and *Mesorhizobium*) (Figure 6A). For the case of HC samples, bacteria of phylum Actinobacteria (*Lawsonella*) correlated positively with the detection of SARS-CoV-2 by RT-ddPCR, while there was also a positive correlation of SARS-CoV-2 detection by RT-qPCR in nasopharyngeal swabs collected from the mothers with members of the phyla Proteobacteria (*Enterobacter*, *Comamonas, Pseudomonas*) and Firmicutes (*Gemella*). There was, in addition, an interesting positive correlation of gestational weeks with members of the phylum Firmicutes (*Blautia*) and a negative correlation of blood glucose levels with Firmicutes (Oscillospiraceae *UCG-002*), Bacteroidetes (*Bacteroides*), and Proteobacteria (*Aeromonas*) (Figure 6B). Finally, the NRS samples showed a negative correlation of SARS-CoV-2 detection by RT-ddPCR with Actinobacteria (*Brevibacterium*), an interesting positive correlation of members of phylum Deinococcota (*Meiothermus*) with the delivery mode (vaginal), and correlation of members of the phyla Firmicutes (*Subdoligranulum*, *Paraclostridium*) and Proteobacteria (*Sphingomonas*) with the male sex of newborns (Figure 6C).

## 3. Discussion

COVID-19, caused by SARS-CoV-2, affects the health of people worldwide, severely affecting adults, and has caused changes in social habits in the population, with priorities placed on prevention and the care of the elderly. In addition, the pandemic has also raised concerns for pregnant women and newborns, and prevention measures were increased in the hospitals [18]. During pregnancy, intestinal dysbiosis associated with COVID-19 inflammation could affect the composition of mothers’ and newborns’ pioneering bacterial communities [19], through the entero-mammary pathway and vertical transmission to the neonate. During COVID-19, there is an interaction between the gastrointestinal and respiratory tract that might cause changes in the gut microbiota. In fact, previous studies have observed alterations in the host microbiota after viral lung infections, resulting in increases in the Bacteroidetes and Firmicutes ratio [20]. Specifically, for COVID-19 patients, significantly lower bacterial diversity and higher relative abundance of opportunistic pathogens have been reported [21].

In our work, upon admission to the hospital, 28 of 33 women with MRS positive for SARS-CoV-2 by RT-ddPCR (MRS+) were asymptomatic, while five had symptoms related to COVID-19 disease; for the group of 17 women with negative MRS, only one exhibited SARS-CoV-2 symptoms. The observed symptoms were breathlessness, cough, diaphoresis, diarrhea, fever, headache, and runny nose, which were lower in comparison to other reports. In a study in the UK population, 342 of 424 (80%) women in the third trimester of pregnancy or peripartum had symptoms on hospital admission, with fever, cough, and breathlessness the most commonly reported [22]. However, pregnant women with COVID-19 were less likely to be symptomatic than non-pregnant counterparts because the risk factors for severe disease include being overweight or obese, older than 35 years of age, and having pre-existing comorbidities [2]. We found a positive trend in SARS-CoV-2 RNA as detected by both tests (RT-qPCR and RT-ddPCR) in male compared to female newborns (Table 2). In other studies, the ratio of Indian male to female infected newborns was 2:1, as was observed in our study [23]. A review of different studies of newborns of different nationalities positive for SARS-CoV-2 reported a male-to-female ratio of 2.8, indicating that male infants were more susceptible to the viral infection than females [24]. The basis of male susceptibility might be explained by the results obtained in murine models, where males are more susceptible to the infection due to the viral recognition of the androgen receptor present in male mice [25]. On the other hand, possible reasons for the differences in SARS-CoV-2 detection among the MRS, HC, and NRS samples might be due to the pathophysiology of the disease, which proceeds from the upper respiratory tract to the lungs, reaching other organs of the lower body including epithelial tissue in the intestinal tract [26]. In addition, differences in viral detection might be associated with the type of sample. A report on SARS-CoV-2 detection using RT-PCR in different types of clinical specimens showed that the rectal swab had a higher positive rate of detection compared to the nasopharyngeal swab, which had a moderate detection rate. The use of rectal swab sampling is recommended for clinical diagnosis of COVID-19 and can be considered a representative gastrointestinal specimen [27]. Additionally, although detection with the RT-qPCR technique is possible with at least 10 genomic copies, more accurate quantification of lower concentrations is obtained via digital PCR, which is based on multiple reactions [28]. All of these might explain the different results for viral detection in the samples of our study, including a higher number of SARS-CoV-2 positive MRS and NRS samples compared to nasopharyngeal swab samples.

The present study reported differences between the microbial composition of SARS-CoV-2 positive versus negative MRS, NRS, and HC samples, analyzed by RT-ddPCR test, obtained from Mexican pregnant women and their neonates. The most abundant phyla found in the different samples were Actinobacteria, Bacteroidetes, Firmicutes, and Proteobacteria, which represented approximately 97% of the microbial composition in each type of sample regardless of the SARS-CoV-2 positivity; these phyla were similar to those in other reports [5,15,29,30]. In the MRS+ samples, members of the Bifidobacteriaceae and Oscillospiraceae families were enriched, while Microbacteriaceae was decreased with a significant difference, suggesting an alteration in the intestinal microbiota due to the presence of the SARS-CoV-2 virus in infected pregnant women. A previous study of nasopharyngeal swab samples taken from Spanish pregnant women with COVID-19 reported a bacterial diversity at the phylum level similar to that in our work, where the Prevotellacea family had the highest abundance [31]. In another comparative study in the Italian population carried out using rectal swab samples from healthy individuals versus COVID-19 patients at different phases of the disease, the comparison of the microbial composition pointed out significant changes in the bacterial communities during progressive phases of the disease [29].

Further, in our work, three representative taxa of the phylum Firmicutes (*Lachnospirraceae*, *Colidextribacter*, *Flavonifractor*) were positively correlated with the presence of SARS-CoV-2 in MRS, and *Faecalibacterium UBA1819* was positively correlated with SARS-CoV-2 detection in nasopharyngeal swabs by Spearman analysis. Interestingly, *Colidexibacter massiliensis* species have been isolated from the human colon of obese patients [32], and an increase in the abundance of members of the families *Ruminococacceae*, *Lachnospiraceae,* and *Enterocacaceae* has been linked to the diagnosis of gestational diabetes mellitus (GDM) [33,34]. On the other hand, *Flavonifractor plautii*, a bacterium involved in the degradation of anticancer flavonoids, has been found in patients with colorectal cancer [35]. The abundance of members of the family *Oscillospiraceae* was significantly increased in the MRS+ samples in our work. Members of the same family as *Faecalibacterium prausnitzzi* present in healthy intestinal microbiota decreased their abundance in the dysbiosis associated with colitis, and it is reported that these bacteria exert an anti-inflammatory action, acting as immunomodulators [5,29,36,37]. A decrease in the abundance of *Faecalibacterium* is associated with GDM [34]; specifically, *Faecalibacterium UBA1819* has been negatively correlated with metabolic syndrome in model mice after dietary fiber supplementation [38]. The genus *Oscillobacter* was found to increase after dietary fiber supplementation in a patient with post-acute COVID-19 syndrome [38], and bacteria of the *NK4A214* group have been linked to low insulin resistance [39]. In other work, authors reported that gut microbiota of a COVID-19 group was dominated by the genera *Streptococcus*, *Rothia*, *Veillonella*, *Eryspelatoclostridium*, and *Actinomyces*, whereas the healthy group was dominated by genera *Romboustia*, *Faecalibacterium*, *Fusicatenibacter*, and *Eubacterium halli* [20]. *Clostridium ramosum*, *Coprobacillus*, and *Clostridium hathewayi* correlated with COVID-19 severity, while *Faecalibacterium prausnitzii* was negatively correlated with disease severity. *Bacteroides thetaiotaomicron*, *B. massiliensis*, *B. dorei*, and *B. ovatus* were inversely correlated with SARS-CoV-2 in fecal samples from patients [20,30].

Moreover, Oscillospiraceae *UCG-010*, Oscillospiraceae *UCG-005*, and Anaerotruncus have been found as potential biomarkers of inflammation [40,41,42], while Oscillospiraceae *UCG-002* might play determinant roles in gut microbial community structure and function leading to the development of IgE-mediated food allergy [43]. Some strains of *Akkermansia* (phylum Verrucomicrobia) were associated with lower insulin sensitivity [37]; *A*. *muciniphila* was detected in COVID-19 patients during hospitalization [5]. The genus *Desulfovibrio* (Proteobacteria), which was positively correlated with glucose in our work, is associated with GDM in women with high fat intake in their diet, causing an increase in the abundance of *Parabacteroides*, *Prevotella*, *Escherichia coli*, *Sutterella*, and *Desulfovibrio*, which weaken the intestinal epithelial permeability [37,44].

On the other hand, *Blautia*, which we found to be negatively correlated with BMI, was found depleted in SARS-CoV-2 infection [45] and was associated with reduced inflammatory response in patients with COVID-19 [5]. *Parvimonas*, associated in our study with MRS negative for SARS-CoV-2 by RT-ddPCR, was also found to increase in abundance after SARS-CoV-2 vaccination in oral samples [46]. *Sphingomonas*, associated in our study with nasopharyngeal swabs negative for the virus in the MRS samples, was correlated with altered serum metabolites in symptomatic COVID-19 patients in other studies [47]. The genera *Butyrcimonas* (Odoribacteriaceae family) and *Porphyromonas* (Porphyromonadacea family), associated with negative detection of SARS-CoV-2 in MRS samples, are bacterial producers of butyric, isobutyric, isovaleric, acetic, and propionic acids [48], and are reported to increase along with COVID-19 recovery [49].

Previous studies have reported that the main phyla in human milk are Firmicutes and Proteobacteria, with differences in abundance between them depending on the milk stages (colostrum, transitional milk, or mature milk). The main phyla detected in the colostrum samples of this study were Actinobacteria, Bacteroidetes, Firmicutes, and Proteobacteria, representing 95% of samples negative for SARS-CoV-2 by RT-ddPCR, and 97% of those samples that tested positive, with no significant differences between HC samples positive or negative SARS-CoV-2. Firmicutes was the most abundant phylum in both groups, with more than 50% represented by members of the Staphyloccaceae family in the negative samples, and 40% in the positive samples. In comparison to other studies, similarities and differences have been found in relation to the bacterial profiles of the colostrum samples. In a study of the Mexican population, the most abundant phyla in human milk samples were Actinobacteria, Bacteroidetes, Firmicutes, and Proteobacteria, representing more than 97% of the total; however, unlike our study, the most abundant phyla were Proteobacteria (55%) and Firmicutes (25%), represented by the families Pseudomonadaceae (71%) and Staphylococaceae (82%) [15]. Similarly, in a study carried out using human milk of Caucasian women, most phyla mentioned above were also found, with Proteobacteria the most abundant (67%), followed by Firmicutes (26%) [50]. In a different study using human milk manually collected 1 month after delivery, Firmicutes was found to be the most abundant phylum, followed by the Staphylococaceae family [51]. A report comparing the bacterial population present in human milk and the skin areola found that the most abundant phylum in milk was Proteobacteria, while for the areola it was Firmicutes, to which we could relate the dominance of the phylum Firmicutes in colostrum samples to the presence of skin flakes in the milk [52]. Additionally, in human milk samples obtained 30 days postpartum, Firmicutes (79%) and Proteobacteria (14%) were the most frequent phyla, and the predominant families were Streptococcaceae (50%), Gemellaceae (15%), and Staphylococcaceae (11%) [53]. Finally, in other studies, in human milk, the most abundant taxa at the genus level were *Streptococcus* (16%), *Ralstonia* (5%), and *Staphylococcus* (5%) [51]; or *Streptococcus* (73%) and *Staphylococcus* (10%) [54].

## 4. Materials and Methods

### 4.1. Study Type and Selection of Subjects

This study was transversal, descriptive, and included 50 mother–neonate pairs recruited at General Hospital “Dr. Gustavo Baz Prada” located in the municipality of Ciudad Netzahualcoyotl (19°25′19″ N 99°00′53″ W) at “Instituto Materno Infantil del Estado de Mexico” (IMIEM) located in Toluca-de-Lerdo, Mexico (19°16′02.0″ N 99°39′41.2″ W) during the epidemiological wave between 17 July 2020 and 13 October 2020. Inclusion criteria were Mexican women from 16 to 38 years old in the third trimester of pregnancy, with no hormonal and no antibiotic treatments. Entries with incomplete or inadequate data were excluded. Based on a survey, sociodemographic and clinical information for mothers and neonates were collected (maternal age, gestational age at delivery, delivery mode, sex, age, clinical data, SARS-CoV-2 diagnosis, blood test, and socioeconomic data). The study was approved by the hospital’s Bioethics Committee in Research, with registry number 208C0101110500T_2020-08. All participants consented to the collection of data and signed informed consent in accordance with the Declaration of Helsinki.

### 4.2. Sample Collection

For this study, 250 samples were obtained by medical staff. For SARS-CoV-2 RNA detection by RT-qPCR, 100 pharyngeal and nasopharyngeal swabs of mothers and neonates were collected according to the Standardized Guidelines for Epidemiological and Laboratory Surveillance of COVID-19 of the Health Secretary of the Mexican Government [55,56]. For SARS-CoV-2 RNA RT-ddPCR detection, 50 mother rectal swabs (MRS) and 50 neonate rectal swabs (NRS), were collected at the time of delivery in accordance with the “Method of collecting a rectal swab” [57]. Fifty colostrum or transition milk samples (HC) were collected from each mother between 0–6 days postpartum. The nipple area was cleaned using a sterile swab and water, and approximately 0.1–1 mL of the sample was collected in a sterile tube. All samples were immediately stored at −20 °C and transported to the Molecular Biology laboratory, Laboratorio Estatal de Salud Pública del Estado-de-México (ISEM Secretaría e Instituto de Salud del Estado de México) (19°16′19″ N 99°39′27″ W), for nucleic acid extraction and analysis.

### 4.3. DNA and Viral Nucleic Acid Extraction

The DNA and viral nucleic acid extraction were performed with 140 µL of nasopharyngeal and pharyngeal swabs in viral transportation medium (VTM), 140 µL fat free HC, 140 µL MRS or NRS in 0.1%PBS using the MagNA Pure 96 and viral MA small volume kit (Cat 06543588001, Roche, Pleasanton, CA, USA). A volume of 260 µL lysis buffer was added to the sample, and the mixture was placed in the cartridge along with other kit reagents in the MagNA Pure 96 Instrument equipment (Pleasanton, Roche, Pleasanton, CA, USA). The HP RNA program Blood_external_lysis DNA and viral nucleic acid small volume purification protocols were used. Extracted nucleic acids were stored at −80 °C and transported to the Laboratory of Environmental Genomics at Cinvestav-Zacatenco (19°30′33″ N 99°07′46″ O) for sequencing.

### 4.4. SARS-CoV-2 Virus Genomic RNA Detection by RT-qPCR

One-Step Real-Time RT-PCR kit (Cat. MAD003941M, Vitro Master Diagnóstica, España) was used, in which the primers and probes are designed for detection of the SARS-CoV-2 E gene (E_Sarbenco_F: 5′-ACA GGT ACG TTA ATA GTT AAT AGC GT−3′, E_Sarbeco_P1: FAM-ACA CTA GCC ATC CTT ACT GCG CTT CG-BBQ, E_Sarbeco_R: 5′-ATA TTG CAG CAG TAC GCA CAC A−3′); and N gene (N_Sarbeco_F: 5′-CAC ATT GGC ACC CGC AAT C−3′, N_Sarbeco_P: FAM-ACT TCC TCA AGG AAC AAC ATT GCC A-BBQ, N_Sarbeco_R: 5′-GAG GAA CGA GAA GAG GCT TG−3′) and RNAse P as housekeeping [58]. The RT-qPCR reaction mixture was in 20 µL final volume (12 µL SARS-CoV-2 MMix plus 8 µL total extracted RNA). The reverse transcription reaction (5 min, 25 °C; 20 min, 50 °C) was followed by 5 min at 95 °C for enzyme deactivation. The PCR cycling program consisted of 45 cycles (30 s, 95 °C for denaturation, and 60 s, 60 °C for alignment and extension) in the CFX96 Touch Real-Time PCR Detection System (BIO-RAD, Hercules, CA, USA). A negative reaction and positive controls of non-infectious synthetic DNA contained in the kit (PC SARS-CoV-2) were used [1].

### 4.5. SARS-CoV-2 Virus Genomic RNA Detection by Digital Droplet PCR (RT-ddPCR)

RT-ddPCR was used to determine the viral genomic load in MRS, HC, and NRS samples, using the One-Step RT-ddPCR kit for probes (Cat. 1864021, BIO-RAD, Hercules, CA, USA). The RT-ddPCR reaction volume was in 20µL final volume (5 µL of 4× of Supermix; 2 µL of 25 U/µL reverse transcriptase enzyme; 1 µL 300 mM DDT; 0.5 µL 100 nM primers and probes targeting the N gene of SARS-CoV-2 [58]; 3 µL of total RNA; and 8.5 µL of H_2_O). Once mixed, the reactants were transferred to the GCR96 cartridge (Cat. 12006858, BIO-RAD, Hercules, CA, USA) and sealed using the PX1 PCR Plate Sealer (Cat. 1814000, BIO-RAD, Hercules, CA, USA). The RT-ddPCR cycling program consisted of 20 min, 50 °C for reverse transcription, followed by 5 min, 95 °C for enzyme deactivation. The cycling program consisted of 45 cycles (30 s, 95 °C for denaturation, and 1 min, 60 °C for alignment and extension) in the QX ONE Droplet Digital PCR System (BIO-RAD, Hercules, CA, USA).

### 4.6. Preparation of the V3 16S rRNA Gene Library

DNA obtained from MRS, HC, and NRS samples was quantified (NanoDrop 2000 Spectrophotometer, Thermo Scientific, Waltham, MA, USA), and ~281 bp amplicon containing the V3 hypervariable region of the 16S rRNA gene was amplified using V3-341F series forward primers (barcode set 1–100) complementary to positions 340–356 segment of *Escherichia coli rrnB* 16S rRNA gene, and V3−518R reverse primer complementary to positions 517–533 of the same molecule (Appendix A) [15,59]. The PCR was conducted in 20 µL final volume reaction (1X HF Buffer (1.5 mM Mg++), 200 µM dNTPs, 0.3 µM each forward and reverse primers, 1–5 ng template DNA, 0.02 U/µL Phusion High-Fidelity DNA Polymerase (Thermo Scientific Baltics UAB, Vilnius, Lithuania), and deionized H_2_O). Each primer had a previous heat treatment (95 °C, 3 min followed by 4 °C, 3 min). The PCR cycling program consisted of 6 min, 98 °C of denaturing, followed by 25 thermo-cycles (12 s, 98 °C; 15 s, 62 °C; 10 s, 72 °C), and 5 min, 72 °C extension in a 720 Thermal Cycler (Applied Biosystems, Singapore). For the positive control DNA, a mixture of identified probiotics was used (Bioleven, Wilmington, DE, USA). Simulated DNA extractions were used as negative controls; in these cases, the ~281 pb amplicon was not observed and consequently not sequenced. Finally, the concentration of every single 1–100 barcoded amplicon was determined, normalized, and pooled to build the final library.

### 4.7. High-Throughput DNA Sequencing

For sequencing, V3-16S rRNA gene pooled library was purified using 2% E-Gel^®^ EX (Cat. G661012, Thermo Scientific, Waltham, MA, USA) in E-Gel iBase Power System (Invitrogen, Israel). The size and concentration of every single library were confirmed using the Agilent 2100 Bioanalyzer System and High Sensitivity DNA kit (Cat. 5067−4626, Agilent, Santa Clara, CA, USA). High-throughput sequencing was performed using Ion OneTouch^™^ 2 Instrument (Life Technologies, Carlsbad, CA, USA), Ion PMG^™^ Hi-Q^™^ View sequencing kit (Cat. A30044, Thermo Fisher Scientific, Waltham, MA, USA), Ion PM^™^ Hi-Q^™^ View OT2 kit (Cat. A29900, Ion 318^™^ Chip kit v2 BC, and Ion Torrent PGM System (Cat. 4488146, Life Technologies, Carlsbad, CA, USA) [15]. Ion Torrent PGM software, Torrent Suite software v4.0.2 was used to demultiplex the sequence data based on their barcodes, and reads were automatically filtered to exclude low-quality (quality score ≤ 20), polyclonal sequences (homopolymers > 6) and errors in primers or barcodes. Filtered data were exported as FASTQ files.

### 4.8. ASV Determination and Taxonomic Annotation

QIIME 2022.2 [60] was used for ASV determination and taxonomic annotation. ASVs were determined with qiime dada2 denoise-single plugin; the option -p-trunc len 170 was used to trim and filter the sequences at this value. Taxonomic annotation was performed using the QIIME 2022.2 [60] feature-classifier classify-consensus-blast plugin, with a percentage of identity of 97%. Silva 138 database (accessed in July 2022) was used with the weighted pre-trained classifier (Weighted Silva 138, 99% OTUs full-length sequences) [61]. Run commands are provided in Appendix A QIIME2 workflow.md.

### 4.9. Bacterial Relative Abundance and Diversity Analyses

For further analyses, R 4.2.0 [62] in RStudio 2022.02.32+492 [63] was used. Phyloseq 1.4.0 [64] package was used for analyses of microbial communities; for instance, alpha diversity Observed species, Chao1, Shannon, and Simpson indexes were calculated, and beta diversity was assessed with Unifrac distance. Qiime artifacts were imported using qiime2R 0.99.6 [65] package. Tidyverse 1.3.1 [66] and dplyr 1.09 [67] packages were used for data frame manipulation. Figures were elaborated with ggplot2 3.3.6 [68], ggpbur 0.4.4 [69], scales 1.2.0 [70] gridExtra 2.3 [71] and ggplotify 0.1.0 [72]. For heatmap elaboration DESEq2 1.3.6 [73] and ComplexHeatmap 2.12.0, were used [74]. Vegan 2.6−2 package [75] was employed for the analysis of similarities (ANOSIM) of beta diversity. Venn diagram and Spearman correlation were performed using microbiome 1.18.0 [76] and eulerr 6.1.1 [77] packages. The full script is provided in the Appendix A phyloseq workflow.R.

### 4.10. SourceTracker 2 and LEfSe Analyses

SourceTracker analysis was conducted with sourcetracker2 giibs plugin [78]; rarefaction depth of 950 samples and p-no-loo parameter were utilized. Linear discriminant analysis effect size (LEfSe) was carried out with the Galaxy module from The Huttenhower Lab [79] using the microbial abundance calculated by QIIME2.

### 4.11. Statistical Analyses

Epidemiological, clinical, and biochemical variables were analyzed using descriptive statistics of the groups. Student’s *t*-test was applied to assess the differences between the metadata, and microbial data of the given groups (SPSS Statistics, version 25). Data were represented with the mean ± SD. Statistics with *p*-values < 0.05 were considered significant. Spearman’s rank correlation coefficient was used to assess the relationship between sample metadata and microbial relative abundance. Differences in beta diversity between groups were assessed with an analysis of similarities (ANOSIM). Differential gene expression analysis based on the negative binomial distribution analysis (DESeq2) was used to highlight relevant taxa in the groups. Data were processed with R, as described above.

### 4.12. Sequence Accession Numbers

The sequence FASTQ files and the corresponding mapping file for all samples used in this study were deposited in the NCBI BioProject ID PRJNA856971 Link https://www.ncbi.nlm.nih.gov/sra/PRJNA856971 (accessed on 8 August 2022).

## 5. Conclusions

The results of our work permit the conclusion that defined changes in the bacterial microbiota diversity are associated with the presence of SARS-CoV-2 in rectal swabs collected from mothers and neonates. In addition, the detection of the virus is also associated with changes in the microbiota present in human milk. Our results and conclusions are valid for the sample; however, they should be interpreted with caution due to the limited number of samples, the sampling method by rectal swab, and the fact that neonates were not breastfed by the mothers during the time that the MRS, NRS and HC samples were collected as a consequence of the hospital protocols due to the COVID-19 pandemic.

## Figures and Tables

**Figure 1 ijms-23-10306-f001:**
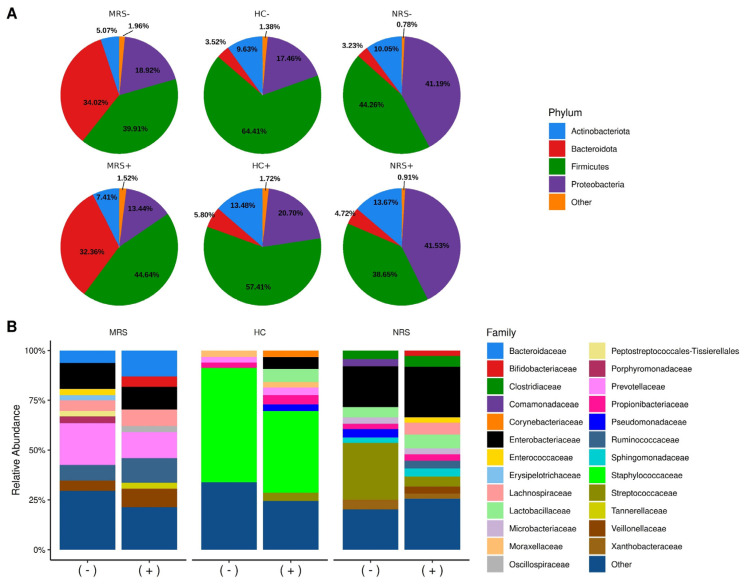
Relative abundances of bacterial taxa in the studied samples. Color sectors indicate taxa as indicated by tags on the right side of the figure; abundances are shown as percentages on the *Y*-axis. Type of sample is shown on top of the figures: MRS (mother rectal swab), HC (human colostrum), NRS (neonate rectal swab), while positive (+) or negative (−) SARS-CoV-2 genome detection by RT-ddPCR is shown at the bottom. (**A**) Pie charts show the top four abundant phyla, while “Other” includes phyla with < 1% relative abundance (Appendix A). (**B**) Bar plots show the top 25 abundant families (Appendix A), while “Other” groups families with < 2.5% relative abundance (Appendix A).

**Figure 2 ijms-23-10306-f002:**
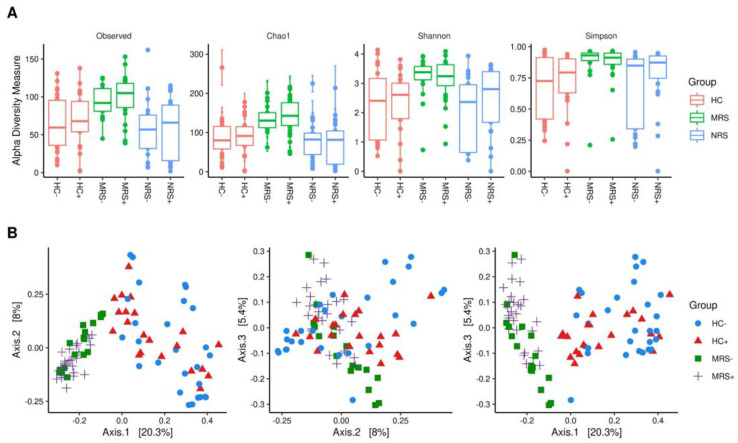
(**A**) Bacterial alpha diversity in the different studied samples. The type of sample is shown on top of graphics as MRS (Mother Rectal Swab), HC (Human Colostrum), NRS (Neonate Rectal Swab), while positive (+) or negative (−) results for SARS-CoV-2 genome detection by RT-ddPCR are shown at the bottom. The Y-axes indicate the values for the Observed number of species, Chao1, Shannon, and Simpson diversity indexes, respectively. Appendix A (**B**) Beta diversity of bacteria in MRS and HC. The graphics show beta-diversity analyses calculated by dissimilarity metrics using features tables and Unweighted UniFrac analysis. The scatter plots were generated using principal coordinates analysis (PCoA) in three different axes showing the percentage of total differences. The positive (+) or negative (−) SARS-CoV-2 genome detection by RT-ddPCR is shown beside the type of labels identified for the samples on the right side of the graphics. MRS and HC differed significantly according to ANOSIM (*p* = 0.001).

**Figure 3 ijms-23-10306-f003:**
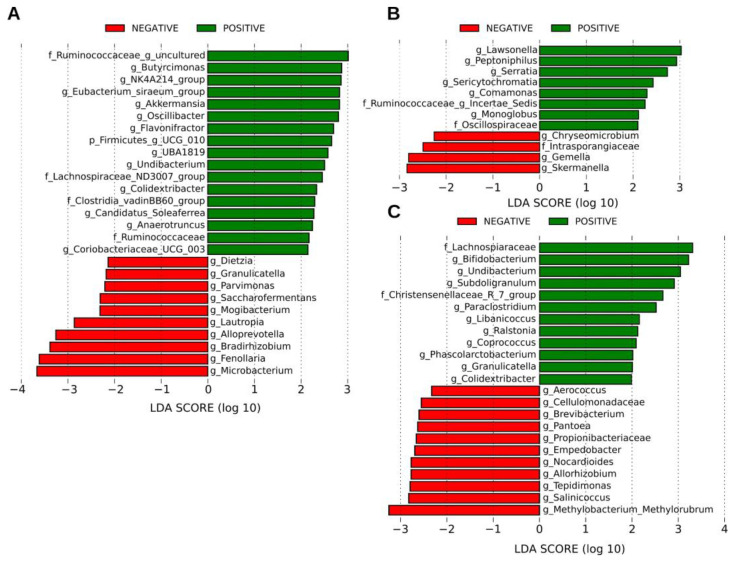
Linear discriminant analysis (LDA) effect size (LEfSe) comparison of differentially abundant bacterial taxa. The figure shows data for (**A**) MRS (mother rectal swab), (**B**) HC (human colostrum), and (**C**) NRS (neonate rectal swab). The effect size for each significant taxon is represented as the log_10_ transformed LDA score shown by the length of the horizontal bars. The threshold on the logarithmic LDA score for discriminative features was set to 2.0. The names of bacterial taxa with a statistically significant change in relative abundance characterizing each condition are written alongside the horizontal lines. Positive (green color) or negative (red color) SARS-CoV-2 genome detection by RT-ddPCR is shown on top of each graphic. Taxa categories are “c”, class; “o”, order; “f”, family, and “g”, genus. See Appendix A for full taxon descriptions and LDA score *p*-values and *q*-values.

**Figure 4 ijms-23-10306-f004:**
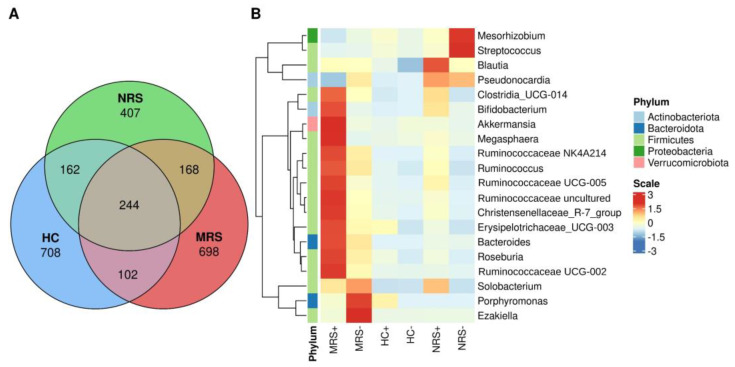
(**A**) Venn diagram showing shared taxa across MRS (mother rectal swab), red color, HC (human colostrum), blue color, NRS (neonate rectal swab), green color. (**B**) Differential frequency of bacteria in the studied samples. The heat-map highlights important taxa among MRS, HC, and NRS, while positive (+) or negative (−) SARS-CoV-2 genome detection by RT-ddPCR is shown beside the label. Columns show the abundance of the 20 bacterial genera with the best *p*-value (*p* < 0.005) calculated by DESeq (differential gene expression based on the negative binomial distribution) and the phyla to which they belong. The color scale from blue (−3) to red (3) indicates the Wald Test coefficient calculated by DESeq from the feature table, with red more frequent. Color keys for phyla are shown on the right side of the figure. Actinobacteriota = Actinobacteria, Bacteroidota = Bacteroidetes.

**Figure 5 ijms-23-10306-f005:**
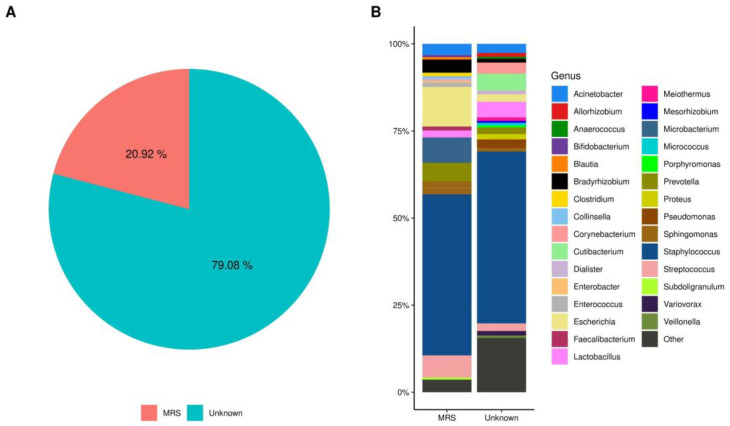
SourceTracker analysis of the possible origin of bacteria in human colostrum. (**A**) Microbial source tracker analysis showing the proportion of bacteria identified in HC (human colostrum) classified by source (*p* < 0.001, Wilcoxon signed-rank test). The red color sector indicates that 20.92% of the bacterial taxa have a probable origin in the MRS (mother rectal swab), while 79.08% have an unknown origin. (**B**) Relative abundance in the percentage of most common bacterial taxa found in the mother’s HC is classified by SourceTracker Gibbs analysis as “MRS and Unknown”. The names of genera are indicated by colored legends on the right side of the graphic (Appendix A).

**Figure 6 ijms-23-10306-f006:**
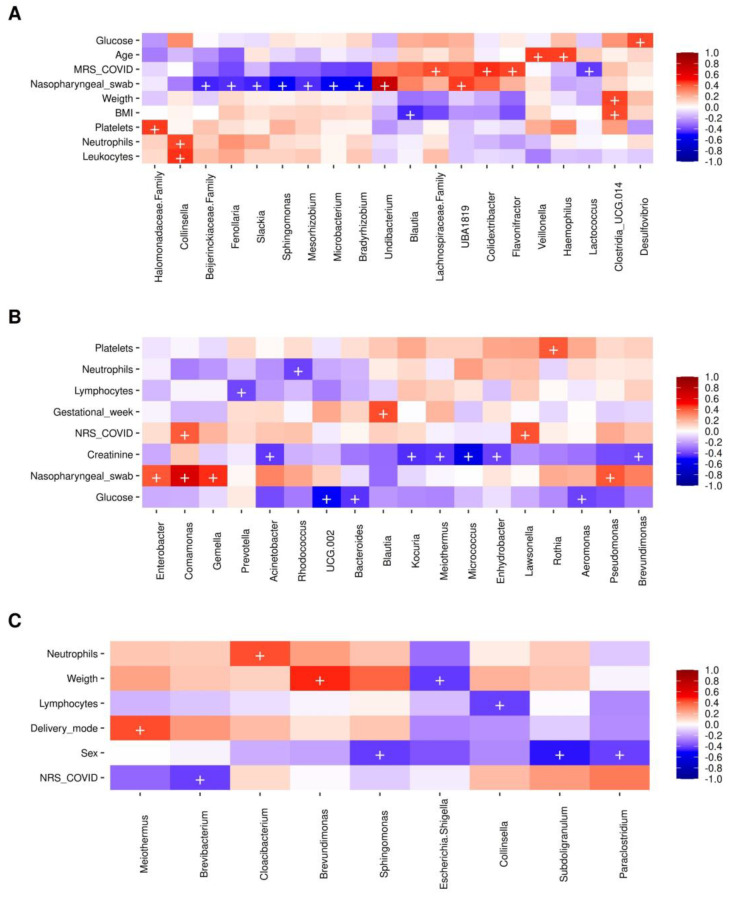
Spearman correlation analysis of clinical metadata and other variables, with bacterial abundance. (**A**) MRS (mother rectal swab); (**B**) HC (human colostrum), and (**C**) NRS (neonate rectal swab). The heatmaps show correlations between different bacterial taxa and numerical metadata. Columns show the bacterial taxa, while rows show the numerical metadata. The color keys from blue (−1, negative) to red (+1, positive) measure the correlation. The plus symbol “+” denotes a significance of *p* < 0.001. For the “Sex” variable, the blue color indicates a correlation of the bacteria with male children; for the “Delivery mode” variable, the red color indicates a correlation with vaginal delivery. Two MRS, two HC, and four NRS samples were not included in these analyses since they lacked suitable metadata.

**Table 1 ijms-23-10306-t001:** Data of 50 participant mothers grouped according to SARS-CoV-2 rectal swab RT-ddPCR detection.

Variable			Positive	Negative	*p*-Value
Number of subjects (*n* = 50)			33 (66%)	17 (34%)	nd
Age (years)			25.48 (±5.19)	25.47 (±6.10)	0.993
Age range			18 to 38	16 to 36	nd
Gestational age (weeks)			38.69 (±1.85)	38.95 (±1.39)	0.571
Weeks range			34 to 41	36.6 to 41.2	nd
Anthropometry		Height (m)	1.57 (±0.05)	1.55 (±0.05)	0.499
		Weight (kg)	74.25 (±13.65)	76.17 (±12.61)	0.624
		Normal (BMI 18.5–24.9)	7 (21.21%)	0 (0.0%)	nd
		Overweight (BMI 25.0–29.9)	10 (30.30%)	9 (52.94%)	nd
		Obesity (BMI > 30.0)	16 (48.48%)	8 (47.06%)	nd
COVID-19 Symptoms		Symptoms	5 (15.15%)	1 (5.88%)	nd
		Asymptomatic	28 (84.84%)	16 (94.11%)	nd
HC samples (*n* = 50)			22 (44.0%)	28 (56.0%)	nd
SARS-CoV-2 nasopharyngeal swab *		Positive	9 (27.3%)	0 (0.0%)	nd
		Negative	24 (72.7%)	17 (100.0%)	nd
Blood test		Reference range			
	Leukocytes (×10^9^/L)	4.5–10	9.17 (±3.31)	8.3 (±2.02)	0.239
	Neutrophils (×10^9^/L)	1.8–8.0	5.97 (±2.69)	5.60 (±1.89)	0.576
	Neutrophils (%)	43.0–65.0	69.29 (±13.86)	66.81 (±14.25)	0.563
	Lymphocytes (×10^9^/L)	1.1–3.2	2.26 (±2.60)	1.7 (±0.59)	0.247
	Lymphocytes (%)	21.0–48.0	23.69 (±20.61)	20.49 (±5.86)	0.520
	Platelet count (×10^9^/L)	150–450	218.68 (±65.89)	225.17 (±49.45)	0.700
	Fasting glucose (mg/dL)	74–06	82.34 (±11.79)	83.35 (±12.32)	0.783
	Creatinine (mg/dL)	0.5–0.9	0.65 (±0.18)	0.66 (±0.11)	0.808
Risk factors		Alcoholism	0	1 (0.17%)	nd
		Smoking	0	1 (0.17%)	nd
Parity	Total		63	40	nd
	Current parities		33	17	nd
		Vaginal	23 (69.69%)	16 (94.11%)	nd
		Cesarean	10 (30.30%)	1 (5.88%)	nd
	Gravity	Uniparous	15 (45.45%)	6 (35.30%)	nd
		Multiparous	18 (54.54%)	11 (64.70%)	nd
	Previous parities		30	23	nd
		Vaginal	18 (72.0%)	8 (34.8%)	nd
		Cesarean	7 (23.3%)	8 (34.8%)	nd
		Abortions	5 (16.7%)	7 (30.4%)	nd
Socioeconomic data					
Educational level		Primary school (6 years)	6 (18.18%)	3 (17.64%)	nd
		Secondary school (3 years)	12 (36.36%)	7 (41.17%)	nd
		High school (3 years)	12 (36.36%)	6 (35.29%)	nd
		University (4–5 years)	2 (6.06%)	1 (5.88%)	nd
		None	1 (3.03%)	0 (0.0%)	nd
Marital status		Free union	24 (48.97%)	9 (52.94%)	nd
		Married	5 (15.15%)	1 (5.88%)	nd
		Single	4 (12.12%)	7 (41.7%)	nd
Main activity		Housewife	28 (84.84%)	14 (82.35%)	nd
		General employees	5 (15.15%)	3 (17.64%)	nd

BMI, body mass index; HC, human colostrum; m, meters; kg, kilograms; mg, milligrams; L, liters. Standard deviation is shown as ± values; *p*-value was calculated according to *t*-test (SPSS version 25.0); *p <* 0.05 is considered statistically significant differences. * SARS-CoV-2 RNA detection by RT-qPCR; nd, not determined.

**Table 2 ijms-23-10306-t002:** Data of 50 participant neonates grouped according to SARS-CoV-2 rectal swab RT-ddPCR detection.

Variable		Positive	Negative	*p*-Value
Number of subjects (*n* = 50)		25 (50%)	25 (50%)	nd
Sex	M	17 (68%)	17 (68%)	nd
	F	8 (32%)	8 (32%)	nd
SARS-CoV-2 nasopharyngeal swab *	Positive	5 (20.0%, M)	0.0 (0.0%)	nd
	Negative	20 (80.0%, F)	25 (100.0%)	nd
Qualification status	Reference range			
APGAR (1 to 10)	6/7	1 (4.0%)	0 (0.0%)	nd
	7/8	1 (4.0%)	0 (0.0%)	nd
	7/9	1 (4.0%)	2 (8.0%)	nd
	8/9	19 (76%)	19 (76%)	nd
	9/9	3 (12%)	4 (16%)	nd
Silverman Andersen	0	20 (80%)	19 (76%)	nd
	0/1	4 (16%)	6 (24%)	nd
	0/3	1 (4.0%)	0 (0.0%)	nd
Capurro (weeks)	Average	38.67 (±1.57)	39.00 (±1.36)	0.430
	Preterm (22–36)	2 (8.0%)	1 (4.0%)	nd
	Term (37–42)	23 (92%)	24 (96%)	nd
Somatometry	Reference range			
Macrosomia (g)	>4000	0 (0.0%)	0 (0.0%)	nd
Proper weight (g)	(2500 to 4000)	24 (96%)	24 (96%)	nd
Low weight (g)	≤2500	1 (4.0%)	1 (4.0%)	nd
IUGR		1 (4.0%)	0 (0.0%)	nd
Weight (kg)		2.88(±0.43)	2.79(±0.41)	0.508
Weight range (kg)		1.66–3.68	2.12–3.257	nd
Size (cm)		49.36 (±1.77)	49.66 (±2.68)	0.644
Size range (cm)		45–52	40–53	nd
Cephalic perimeter (cm)		33 (±1.36)	33 (±1.32)	0.526
Abdominal perimeter (cm)		31 (±2.04)	31 (±2.19)	0.812
Perinatal asphyxiation		1 (4.0%)	0 (0.0%)	nd
Breathing difficulty		4 (16%)	0 (0.0%)	nd
Infection		1 (4.0%)	0 (0.0%)	nd

APGAR (Appearance, Pulse, Grimace, Activity, and Respiration); IUGR (Intrauterine Growth Restriction); M (Male); F (Female). Standard deviation is shown as ±values; *p-*value was calculated according to *t*-test (SPSS version 25.0); *p-*value was calculated for categorical data; *p* < 0.05 are considered statistically significant differences; Days of birth ≤ 6. * SARS-CoV-2 RNA detection by RT-qPCR; kg, kilograms; g, grams; cm, centimeters; nd, not determined.

**Table 3 ijms-23-10306-t003:** Sequencing summary for all samples of this study.

	MRS (*n* = 45)	HC (*n* = 50)	NRS (*n* = 49)
Parameter before trimming			
Total forward reads	2,451,400.00	1,617,526.00	2,188,670.00
Forward reads mean	54,475.55	32,350.52	44,666.00
Min-Max forward reads	9241.00–183,829.00	6557.00–231,169.00	2723.00–240,577.00
Sequence length (median)	193	198	198
Samples with <10,000 reads	1	10	19
Parameter after trimming			
QS (median)	32	32	30
Percentage of identity (97%)			
Total ASV counts	2052	2028	1560
Identified ASVs	346	475	371

MRS, Mother Rectal Swab; HC, Human Colostrum; NRS, Neonate Rectal Swab; QS, Quality Score, Trimmed less than 170; ASVs, Amplicon Sequencing Variants, *n*, number of samples in the category.

## Data Availability

The sequence FASTQ files, and the corresponding mapping file for all samples used in this study, were deposited in the NCBI BioProject ID PRJNA856971 Link https://www.ncbi.nlm.nih.gov/sra/PRJNA856971 (accessed on 8 August 2022).

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
