# Peer review of "The Entero-Mammary Pathway and Perinatal Transmission of Gut Microbiota and SARS-CoV-2"

_ijms, 2022, doi:10.3390/ijms231810306_

Round 1

Reviewer 1 Report

Authors demonstrated that defined changes in the bacterial microbiota diversity are associated to the presence of SARS-CoV-2 in rectal swabs collected from mothers and neonates. In addition, the detection of the virus is also associated to changes in the microbiota present in the human milk. 

1. Line 111-118, RT-ddPCR showed differences in positive results for the virus in different groups, and possible reasons should be discussed.

2. A discussion of the clinical significance of these bacterial differences is suggested in the Discussion

3. Are any of these bacteria selected for further analysis in vitro?

Author Response

Manuscript ID: ijms-1884082

Title: The Entero-Mammary Pathway and Perinatal Transmission of Gut Microbiota and SARS-CoV-2

Authors: Carmen Josefina Juárez-Castelán, Juan Manuel Vélez-Ixta, Karina Corona-Cervantes, Alberto Piña-Escobedo, Yair Cruz-Narváez, Alejandro Hinojosa-Velasco, María Esther Landero-Montes-de-Oca, Eduardo Davila-González, Eduardo González-del Olmo, Fernando Bastida-González, Paola Berenice Zárate-Segura, Jaime García-Mena

Reviewer 1 (Round 1)

Comments and Suggestions for Authors

Authors demonstrated that defined changes in the bacterial microbiota diversity are associated to the presence of SARS-CoV-2 in rectal swabs collected from mothers and neonates. In addition, the detection of the virus is also associated to changes in the microbiota present in the human milk.

  1. Line 111-118, RT-ddPCR showed differences in positive results for the virus in different groups, and possible reasons should be discussed.

Answer: we agree with the reviewer, and to improve the information in “2.1. Presence of SARS-CoV-2 genomic RNA in the mother-neonate pairs” of the Results section, we amended the text of the entire section. In addition, we added the following paragraph to the “Discussion” section: “We found a positive trend in SARS-CoV-2 RNA as detected by both tests (RT-qPCR and RT-ddPCR) in male compared to female newborns (Table 2). In other studies, the ratio of Indian male to female infected newborns was 2:1, as was observed in our study (Kalamidani et al., 2020). A review of different studies of newborns positive for SARS-CoV-2 of different nationalities reported a male-to-female ratio of 2.8, indicating that male infants were more susceptible to the viral infection than females (De Bernardo et al., 2020). The basis of the male susceptibility might be explained by the results obtained in murine models, where males are more susceptible to the infection due to the viral recognition of the androgens receptor present in male mice (Channappanavar et at., 2017). On the other hand, possible reasons for the differences in SARS-CoV-2 detection among the MRS, HC, and NRS samples, might be due to the pathophysiology of the disease which proceeds from the upper respiratory tract to the lungs reaching other organs of the lower body including epithelial tissue in the intestinal tract (Sridhar and Nicholls, 2021). In addition, differences in viral detection might be associated with the type of sample. A report of SARS-CoV-2 detection using RT-PCR in different types of clinical specimens; showed that the rectal swab had a higher positive rate of detection compared to the nasopharyngeal swab which has a moderate detection. The use of rectal swab sampling is recommended for clinical diagnosis of COVID-19 and can be considered a representative gastrointestinal specimen (Bwire et al., 2021). Also, although detection with the RT-qPCR technique is possible with at least 10 genomic copies, more accurate quantification of lower concentrations is obtained via digital PCR, which is based on multiple reactions (Bustin et al., 2009). All of these might explain the different results in the viral detection in samples of our study, including a higher number of SARS-CoV-2 positive cases in MRS and NRS, compared to nasopharyngeal swab samples”, please see lines 328-350.

  1. A discussion of the clinical significance of these bacterial differences is suggested in the Discussion

Answer: we thank the reviewer for the opportunity to improve the Discussion section of our manuscript. We reviewed the whole discussion and added new information as follows: “In our work, upon admission to the hospital, 28 of 33 women with positive MRS to SARS-CoV-2 by RT-ddPCR (MRS+) were asymptomatic, while five had symptoms related to COVID-19 disease; for the group of 17 women with negative MRS, only one exhibited SARS-CoV-2 symptoms” lines 318-321. The clinical significance of the difference in the bacterial community diversity found in the samples, has been specified in lines 360-362, 365-368, 370-384, 395-404, 405-407 where we point out the association of several bacterial taxa with the COVID-19 severity and the presence of other diseases.

  1. Are any of these bacteria selected for further analysis in vitro?

Answer: we thank the reviewer for the interesting suggestion; we are currently working on isolation assays procedures.

References for Reviewer 1

Bustin, S.A.; Benes, V.; Garson, J.A.; Hellemans, J.; Huggett, J.; Kubista, M.; Mueller, R.; Nolan, T.; Pfaffl, M.W.; Shipley, G.L.; et al. The MIQE Guidelines: Minimum Information for Publication of Quantitative Real-Time PCR Experiments. Clin Chem 2009, 55, 611–622, doi:10.1373/CLINCHEM.2008.112797.

Bwire, G.M.; Majigo, M. v.; Njiro, B.J.; Mawazo, A. Detection Profile of SARS-CoV-2 Using RT-PCR in Different Types of Clinical Specimens: A Systematic Review and Meta-Analysis. J Med Virol 2021, 93, 719–725, doi:10.1002/JMV.26349.

Channappanavar, R.; Fett, C.; Mack, M.; ten Eyck, P.P.; Meyerholz, D.K.; Perlman, S. Sex-Based Differences in Susceptibility to SARS-CoV Infection. J Immunol 2017, 198, 4046, doi:10.4049/JIMMUNOL.1601896.

de Bernardo, G.; Giordano, M.; Zollo, G.; Chiatto, F.; Sordino, D.; de Santis, R.; Perrone, S. The Clinical Course of SARS-CoV-2 Positive Neonates. Journal of Perinatology 2020, 40, 1462, doi:10.1038/S41372-020-0715-0.

Kalamdani, P.; Kalathingal, T.; Manerkar, S.; Mondkar, J. Clinical Profile of SARS-CoV-2 Infected Neonates From a Tertiary Government Hospital in Mumbai, India. Indian Pediatr 2020, 57, 1143, doi:10.1007/S13312-020-2070-9.

Sridhar S, Nicholls J. Pathophysiology of infection with SARS‐CoV‐2—What is known and what remains a mystery. Respirology. 2021;26:652–665. 10.1111/resp.14091

---end-of-text---

Reviewer 2 Report

The authors report a transversal study that analyzes the influence of the intestinal microbiota in women with SARS-CoV-2. This study is in detail characterization of the entero-mammary microbiota of women in the presence of the virus during childbirth. The manuscripts appear to be very interesting. All paragraphs are described in detail, the results are understandable, and the manuscript may be accepted in its current form, after the following minor revisions:

Introduction

- Line 45: the authors should include statistical studies documenting the incidence of pregnant women with SARS-CoV 2 during these pandemic years.

- Line 53: how does it affect the functionality of the intestinal microbiota? Report some studies to explain in more detail.

Materials and methods:

-Line 395: Study type and selection of subjects. The authors could include, in materials and methods, a summary table of the main characteristics of the subjects included in the study.

-Line 436: 4.4. SARS-CoV-2 virus genomic RNA detection by RT-qPCR. Report the forward and reverse sequences used for the each genes

Results

-In Figure 1, the authors could use pie charts to describe the relative abundances of the phyla and identify the percentage of each phylum

Discussions

The discussion paragraph should be revised, the authors should better comment on the results obtained on the basis of literature studies.

Author Response

Manuscript ID: ijms-1884082

Title: The Entero-Mammary Pathway and Perinatal Transmission of Gut Microbiota and SARS-CoV-2

Authors: Carmen Josefina Juárez-Castelán, Juan Manuel Vélez-Ixta, Karina Corona-Cervantes, Alberto Piña-Escobedo, Yair Cruz-Narváez, Alejandro Hinojosa-Velasco, María Esther Landero-Montes-de-Oca, Eduardo Davila-González, Eduardo González-del Olmo, Fernando Bastida-González, Paola Berenice Zárate-Segura, Jaime García-Mena

Reviewer 2 (Round 1)

Comments and Suggestions for Authors

The authors report a transversal study that analyzes the influence of the intestinal microbiota in women with SARS-CoV-2. This study is in detail characterization of the entero-mammary microbiota of women in the presence of the virus during childbirth. The manuscripts appear to be very interesting. All paragraphs are described in detail, the results are understandable, and the manuscript may be accepted in its current form, after the following minor revisions:

Introduction section: (2)

  1. Line 45: the authors should include statistical studies documenting the incidence of pregnant women with SARS-CoV 2 during these pandemic years.

Answer: we thank the reviewer for the suggestion to improve the Introduction; we added more information and references on this matter: “For this matter, statistics from the UK report a total of 9% of pregnant women and 6-week postpartum COVID-19 admissions to intensive care units (ICU) [Nana et al., 2021], while a review highlighted that most infections occurred during the third trimester, 11% of pregnant women with COVID-19 required admission to ICU, and 8% required mechanical ventilation [Mark et al, 2021]. It has also been reported that prevalence among women is variable, rising from 0.5% to 5% in the span of two weeks [Campbell et al., 2020]. Regarding the Mexican population, it was reported that during the second peak of the pandemic, approximately 12% of asymptomatic pregnant women were positive to the virus [Ramírez-Rosas et al, 2021]” lines 47-54.

  1. Line 53: how does it affect the functionality of the intestinal microbiota? Report some studies to explain in more detail.

Answer: we agree with the reviewer’s suggestion; we believe there are changes in the microbiota’s diversity due to the inflammatory process, consequently some non-desirable taxa (pathogens) proliferate changing the microbiota’s performance. We added the following text and references on this matter:

In lines 311-317 “During COVID-19, there is an interaction between the gastrointestinal and respiratory tract which might cause changes in the gut microbiota. In fact, previous studies have observed alterations in the host microbiota after viral lung infections, resulting in increases in Bacteroidetes and Firmicutes ratio (Li et al., 2020). Specifically, for COVID-19 patients, significantly lower bacterial diversity and higher relative abundance of opportunistic pathogens have been reported (Marsland et al., 2015).”

In lines 385-392. “In other work, authors report that gut microbiota of COVID-19 group was dominated by the genera Streptococcus, Rothia, Veillonella, Eryspelatoclostridium, and Actinomyces, whereas the healthy group was dominated by genera Romboustia, Faecalibacterium, Fusicatenibacter, and Eubacterium halli (Li et al., 2020). Clostridium ramosum, Coprobacillus, and Clostridium hathewayi correlated with COVID-19 severity, while Faecalibacterium prausnitzii was negatively correlated with disease severity. Bacteroides thetaiotaomicron, B. massiliensis, B. dorei, and B. ovatus were inversely correlated with SARS-CoV-2 in fecal samples from patients (Li et al., 2020; Zuo et al., 2020).”

Materials and methods section: (2)

  1. Line 395: Study type and selection of subjects. The authors could include, in materials and methods, a summary table of the main characteristics of the subjects included in the study.

Answer: we thank the reviewer for getting this issue to our attention, however, the suggested information is already indicated in lines 100-104, and Table 1 and Table2. However, for the sake of clarity, we added the following information in lines 104-107: “The fifty neonates included in the study were less than 6 days old, 32% of them were female and 68% were male, with an average weight of 2.92 (±0.42) kg and size of 49.51 (±1.47) cm

  1. Line 436: 4.4. SARS-CoV-2 virus genomic RNA detection by RT-qPCR. Report the forward and reverse sequences used for the each genes

Answer: the reviewer’s suggestion was honored, and we added the requested primer sequences lines 486-490 in “4.4. SARS-CoV-2 virus genomic RNA detection by RT-qPCR” of “4. Materials and Methods” section: “(E_Sarbenco_F: 5’-ACA GGT ACG TTA ATA GTT AAT AGC GT-3’, E_Sarbeco_P1: FAM-ACA CTA GCC ATC CTT ACT GCG CTT CG-BBQ, E_Sarbeco_R: 5’-ATA TTG CAG CAG TAC GCA CAC A-3’); and N gene (N_Sarbeco_F: 5’-CAC ATT GGC ACC CGC AAT C-3’, N_Sarbeco_P: FAM-ACT TCC TCA AGG AAC AAC ATT GCC A-BBQ, N_Sarbeco_R: 5’-GAG GAA CGA GAA GAG GCT TG-3’) and RNAse P as housekeeping”.

Results section (1)

  1. In Figure 1, the authors could use pie charts to describe the relative abundances of the phyla and identify the percentage of each phylum

Answer: in accordance with the reviewer suggestion, the Figure 1 was amended, and the percentage of relative abundances are shown in pie charts along with the percentage of each phylum. The corresponding legend was also modified.

Discussions section (1)

  1. The discussion paragraph should be revised, the authors should better comment on the results obtained on the basis of literature studies.

Answer: we thank the reviewer suggestion that allows us to improve our Discussion section. New comments were added in lines 328-350, 318-321, 311-317, 385-392.

References for Reviewer 2

Campbell, K.H.; Tornatore, J.M.; Lawrence, K.E.; Illuzzi, J.L.; Sussman, L.S.; Lipkind, H.S.; Pettker, C.M. Prevalence of SARS-CoV-2 among Patients Admitted for Childbirth in Southern Connecticut. JAMA - Journal of the American Medical Association 2020, 323, 2520–2522.

Li, F.; Lu, H.; Li, X.; Wang, X.; Zhang, Q.; Mi, L. The Impact of COVID-19 on Intestinal Flora: A Protocol for Systematic Review and Meta Analysis. Medicine 2020, 99, e22273, doi:10.1097/MD.0000000000022273.

Mark, E.G.; McAleese, S.; Golden, W.C.; Gilmore, M.M.; Sick-Samuels, A.; Curless, M.S.; Nogee, L.M.; Milstone, A.M.; Johnson, J. Coronavirus Disease 2019 in Pregnancy and Outcomes among Pregnant Women and Neonates: A Literature Review. Pediatric Infectious Disease Journal 2021, 40, 473–478.

Marsland, B.J.; Trompette, A.; Gollwitzer, E.S. The Gut-Lung Axis in Respiratory Disease. Ann Am Thorac Soc 2015, 12 Suppl 2, S150–S156, doi:10.1513/ANNALSATS.201503-133AW.

Nana, M.; Nelson-Piercy, C. COVID-19 in Pregnancy. Clin Med (Lond) 2021, 21, E446–E450, doi:10.7861/CLINMED.2021-0503.

Ramírez-Rosas, A.; Benitez-Guerrero, T.; Corona-Cervantes, K.; Vélez-Ixta, J.M.; Zavala-Torres, N.G.; Cuenca-Leija, J.; Martínez-Pichardo, S.; Landero-Montes-de-Oca, M.E.; Bastida-González, F.G.; Zárate-Segura, P.B.; et al. Study of Perinatal Transmission of SARS-CoV-2 in a Mexican Public Hospital. International Journal of Infectious Diseases 2021, 113, 225–232, doi:10.1016/J.IJID.2021.10.006.

Zuo, T.; Zhang, F.; Lui, G.C.Y.; Yeoh, Y.K.; Li, A.Y.L.; Zhan, H.; Wan, Y.; Chung, A.C.K.; Cheung, C.P.; Chen, N.; et al. Alterations in Gut Microbiota of Patients With COVID-19 During Time of Hospitalization. Gastroenterology 2020, 159, 944, doi:10.1053/J.GASTRO.2020.05.048.

---end-of-text---

Round 2

Reviewer 1 Report

In their revised manuscript, the authors have revised sufficiently addressed my concerns.